# VOCs Analysis of Three Different Cultivars of Watermelon (*Citrullus lanatus* L.) Whole Dietary Fiber

**DOI:** 10.3390/molecules27248747

**Published:** 2022-12-09

**Authors:** Laura Maletti, Veronica D’Eusanio, Caterina Durante, Andrea Marchetti, Lorenzo Tassi

**Affiliations:** 1Department of Chemical and Geological Sciences, University of Modena and Reggio Emilia, 41121 Modena, Italy; 2Consorzio Interuniversitario Nazionale per la Scienza e Tecnologia dei Materiali (INSTM), 50121 Firenze, Italy; 3Interdepartmental Research Center BIOGEST-SITEIA, University of Modena and Reggio Emilia, 41121 Reggio Emilia, Italy

**Keywords:** dietary fibers, food waste, HS-SPME-GC-MS, volatile compounds, watermelon, recycle, nutrients recovery, biorefinery

## Abstract

In this study, the trend of VOCs of dietary fiber samples, coming from three different watermelon cultivars *Citrullus lanatus* L. (variety *Gavina^®®^, Crimson Sweet*, and *Asahi Miyako*) was investigated. This foodstuff, obtained as a by-product of residual agri-food production, has gained increasing attention because of its many bioactive components and high dietary fiber content. The result is a fibrous material for specific applications in food manufacturing, such as corrector for some functional and technological properties. In this study, a method based on headspace solid-phase microextraction (HS-SPME) coupled with gas chromatography–mass spectrometry (GC–MS) was used to characterize the aromatic profiles of the dried raw materials. Therefore, the VOCs of the samples of the three cultivars were investigated. Experimental results have shown that watermelon fibers generate VOCs, which can be grouped into six common classes of analytes. The different distributions of the identified compounds made it possible to effectively differentiate the three cultivars studied based on their peculiar aroma profiles. In particular, *Gavina*^®®^ fiber is distinguished by the high content of terpenes, *Asahi Miyako* by the presence of aldehydes generated as fatty acid metabolites, and *Crimson Sweet* by the higher content of acetyl esters.

## 1. Introduction

Watermelon (*Citrullus lanatus* L.) is an annual crop, typical of the summer season, appreciated by many people across the world. Its nutritional potential includes the reduction in the risk of cardiovascular and neurodegenerative diseases, the prevention of metabolic disorders, such as insulin resistance and diabetes, and certain types of cancers [1,2,3]. The beneficial properties of this fruit are attributed to the content of important phytochemical molecules, including lycopene, β-carotene, and polyphenols [4], bioactive species capable of exerting powerful antioxidant and anti-inflammatory effects [2]. Furthermore, it is a precious source of mineral salts, in particular K, Mg, Ca, and Fe [3], and of some vitamins: one cup of diced watermelon covers about 21% of the daily requirement of vitamin C and 17% of vitamin A [1]. Finally, it contains a good amount of dietary fiber, the consumption of which is strongly recommended in the human diet by several well-documented reports, including those of the Food and Agriculture Organization of the United Nations (FAO) [5,6,7,8].

Watermelon cultivation extends over an area that covers about 7% of the total land surface dedicated to the global production of fruit and vegetables [4]. In territories with temperate climates, its commercial availability indicatively lasts from May to September. According to FAOSTAT (FAO Statistics Division) [9], the production of watermelon in Italy increased from ~ 580 Ktons in 2018 to ~ 650 Ktons in 2020, with a cultivated area of about 13,400 ha. The Italian National Institute of Statistics (ISTAT) [10] estimated that during the cultivation cycle, ≈ 8.5% of the total production is not harvested, corresponding to about 55 Ktons/year. These data become impressive if considering the worldwide production of watermelon. In 2017–2018, about 42 million tons of watermelon by-products were generated during the transformation, preparation, and consumption of the fruit [11]. In addition, the quantity of fresh products that annually remains uncollected can reach ~20% of the total production [12]. In fact, fruits with characteristics that are not optimal for consumers, such as shape, size, color, or with defects or damage, are removed from the supply chain during the sorting operations. Therefore, they are left in the cultivation field at the end of the season. Furthermore, watermelon waste biomass is increased by the amount of fruit further discarded or unsold by retailers.

In addition to food loss problem, the disposal of these large biomasses of *Cucurbitaceae* is also a problem. In fact, fruits with high water content and rich in fermentable sugars, if spread on the land, give rise to a large quantity of alcohols, i.e., methanol and ethanol. These species cause serious consequences for the microbial flora of the soil, to the point of even procuring sterility [13]. In this context arises the need to recover the residual products of primary agronomic productions, i.e., the massive quantity of fruit that remains in a pre-waste condition on the crop field, and their precious nutritional components [5,14,15].

In this study, three watermelon pomace whole dietary fibers, WDFs, obtained from three different *Citrullus lanatus* cultivars, *Gavina*^®®^, *Crimson Sweet*, and *Asahi Myiako*, were examined as functional products to be introduced into new food formulations as well as to support the recovery of the watermelon by-product in a circular economy context.

The application of biomaterials in the food industry has significantly increased in recent years. In fact, consumer demand for functional products enriched with value-added components, such as dietary fiber and bioactive molecules beneficial to human health [16,17], is growing sharply. Several studies have shown that food waste can be used as potential added ingredients with high nutritional value and can improve some technological properties in doughs and formulations [18,19,20]. This is the perfect demonstration of the ways in which health claims benefit not only consumers, but also the food industry, always looking for new, cheaper and healthier ingredients and the environment by reducing waste and optimizing resources.

In particular, the volatile fraction of the investigated fibers, rich of a wide range of naturally occurring bioactive compounds, was chemically characterized by HS-SPME-GC-MS. The analysis of the volatile fraction of food is essential for understanding, planning or improving the sensory characteristics of new possible products [21]. Aroma of food is strictly dependent on the chemical composition [22,23] and imparts specific quality features. Indeed, it represents a valuable parameter of great interest, especially in the food industry, since it is directly related to the product’s acceptability by consumers as well as to its nutritional and qualitative characteristics and in terms of healthiness and safety of the product itself [23]. Furthermore, in the specific case of fruits, the volatile organic compounds (VOCs) allow to identify and highlight any differences between different cultivars of the same species [22]. Indeed, the geographical origin, cultivation site, production method, harvesting technology and degree of ripeness significantly affect the chemical composition of the fruits, with consequent modification of their relative aromatic bouquet [23,24].

Furthermore, VOCs analysis of the investigated WDF materials was carried out to identify possible common characters or any analytes that differentiate the fiber of different cultivars. *Gavina*^®®^ variety is a mini watermelon, a seedless hybrid, specifically obtained with excellent organoleptic characteristics and small dimensions. This cultivar is covered with trademark and the commercial brand is clearly identified by the origin from Sardinian territory (Italy). *Asahi Miyako*, instead, contains black seeds, but sometimes these may not ripen sufficiently. In this case, the seeds are white, soft, and small. *Crimson Sweet* is one of the most popular varieties of watermelon. Its seeds are dark in color, similar to those of *Miyako*, but slightly bigger.

## 2. Results

The first aim of this research was to develop a straightforward methodology for the effective characterization of the VOC aromatic profile of dietary fiber recovered from pre-waste watermelon fruits. Furthermore, in this study, we aimed to obtain a suitable screening method for sampling VOCs from the WDF of the pomace of different cultivars of *Citrullus lanatus*. The aromatic and fragrant characteristics of a fruit, in fact, can vary significantly from one cultivar to another [22]. This occurs because the VOCs are produced through metabolic pathways that develop during ripening, harvesting and storage processes, all strongly affected by variety-dependent factors [22,23,24]. The variables relating to the soil where the fruit is grown must also be considered, i.e., the geographical position and the micro-climate of the place. 

Figure 1 shows the watermelon pulps without seeds from the three different cultivars: *Gavina*^®®^, *Asahi Miyako*, *Crimson Sweet*.

The most substantial difference between the three pulps concerns the *Miyako* variety. In fact, it contains some residual seeds that are not completely ripe, which are difficult to further separate from the pulp.

Figure 2 shows the watermelon pomace whole dietary fiber (WDF) immediately after the drying process (left) and after a subsequent grinding process (right). Only WDF of the *Gavina*^®®^ cultivar (G_WDF) is reported as the appearance of the dried samples of the other two cultivars; *Asahi Miyako* (AM_WDF) and *Crimson Sweet* (CS_WDF) are similar.

Headspace profiles obtained from the vacuum freeze-dried samples of WDF *Gavina*^®®^, *Asahi Miyako* and *Crimson Sweet* are shown in Figure 3a–c, respectively. Several compounds were identified on the basis of a combination of some or all of the following criteria: (i) mass spectral data of the libraries supplied with the operating system of the GC-MS and from mass spectra databases; (ii) mass spectra found in the literature; (iii) mass spectra and retention time of an injected standard; (iv) LRI values, typically used for the identification of isomers. The identified compounds are summarized in Table 1. The reproducibility of the results is expressed as standard deviation (SD_3_), where the subscript “three” indicates the replicates number.

From Table 1, it can be noticed that the compounds determined in WDFs are represented by acids, carbonyl compounds, esters, alcohols, terpenes and other compounds. In particular:In G_WDF sample, 93 compounds were detected, including 34 aldehydes (ALD), 15 alcohols (ALC), 1 acetate ester (ACE), 8 ketones (KET), 25 terpenes and derivatives (TER), and 10 other compounds (OTH);In AM_WDF sample, 64 compounds were detected, including 19 aldehydes (ALD), 11 alcohols (ALC), 10 ketones (KET), 1 acetate ester (ACE), 18 terpenes and derivatives (TER), and 5 other compounds (OTH);In CS_WDF specimen, 44 compounds were detected, including 13 aldehydes (ALD), 14 alcohols (ALC), 3 ketones (KET), 1 acetate ester (ACE), 9 terpenes and derivatives (TER), and 4 other compounds (OTH).

Some differences were observed in the VOCs profiles. *Crimson Sweet* WDF chromatogram is the simplest in terms of both the number of peaks and their intensity. On the other hand, *Asahi Miyako* and *Gavina*^®®^-related profiles are more complex, with a greater number of peaks and analytes. Figure 4 shows the number of analytes in each chemical class for each sample. 

## 3. Discussion


*Classification of Volatile Compounds in Watermelon Pomace Fiber*


Although the three WDF samples originate from the same family, *Citrullus lanatus*, it is evident that each aromatic bouquet is strongly influenced by the cultivar. The VOCs are produced through metabolic pathways that develop during the ripening, harvesting and storage processes [22,23,24], all strongly affecting one each other, in addition to cultivar-dependent factors. The variables relating to the soil where the fruit is grown must also be considered, i.e., the geographical site and the microclimate of the place [22,40]. The different compounds are discussed separately based on their specific chemical class.

### 3.1. Aldehydes

As previously mentioned, G_WDF sample has a headspace populated by the highest number of species belonging to this class. However, observing the TIC-based total abundance, it is possible to note that the AM_WDF sample is the one with the greatest value. The *Gavina*^®®^ fiber presents a more complex and diversified aldehyde group of HS analytes population, while the *Asahi Miyako* one is the simplest in number of analytes, but characterized by a higher abundance. The aromatic bouquet of the *Crimson Sweet* sample, however, is scarcely characterized by this class of molecules. Both the number of peaks and their abundance are in fact extremely low for this specimen. 

Previous studies suggested that saturated and unsaturated C9 aliphatic aldehydes, together with C9 alcohols, play a key role in watermelon aroma [22,23,25,30,33,34]. All of these molecules contribute to some extent to impart waxy, fresh and green notes to the smell of watermelon. We have identified 4-nonenal, (Z,Z)-3,6-nonadienal, (Z)-2-nonenal, (E,E)-2,6-nonadienal, (E)-2-nonenal, and 2,4-nonadienal in our samples. These analytes have already been reported in previous studies about analogous matrices [30,35,37]. Among others, they are considered the species that mainly contribute and define the characteristic and pleasant aroma of watermelon. They are produced at a biological level following the enzymatic oxidation of polyunsaturated fatty acids (PUFA), in particular linoleic acids, by lipoxygenase [24,34] (Figure 5).

This radical process involves the formation of C9 hydroperoxides, highly unstable and subjected to degradation. In fact, several studies have reported the correlation between the concentration increase in 2-nonenal and 2,6-nonadienal and the concentration decrease in linoleic and linolenic acids [39]. Only (E)-2-nonenal was detected in the CS_WDF sample, while G_WDF specimen shows the HS particularly rich in C9 aldehydes, which cover about 13% of the entire VOCs fraction TIC area. Therefore, the *Gavina*^®®^ fiber retains the typical aroma of watermelon more and more after the preparation. In fact, its aromatic bouquet is the only one that contains (Z,Z)-3,6-nonadienal. This aldehyde is often cited in the literature, but is not easily identified, because it is a particularly labile molecule, subject to both isomerization and oxidation. It represents the most characteristic aroma of *Citrullus lanatus* due to its low threshold value and strong odor of “freshly cut watermelon”. For this reason, some authors have defined it as “watermelon aldehyde” [37]. However, given its instability, it is scarcely used in the context of the food industry as an added natural flavoring, since it is difficult to preserve. The most common and recurrent hypothesis is that it rapidly isomerizes to the 2,6-nonadienal and 2-nonenal isomers [28,34,45], especially in matrices that have undergone stressful mixing and homogenization processes [34].

The other aldehydes, on the other hand, are considered to contribute to the watermelon aroma to a lesser, but not negligible extent. They are all molecules with very pleasant fragrances that enrich and complete the aroma of the watermelon WDF. In addition to C9 aldehydes, pentanal, hexanal, 2-hexenal, 4-heptenal, 2,4-hexadienal, 2-heptenal, 2,4-heptadienal, decanal, 2,4-decadienal can be generated by the oxidation of polyunsaturated fatty acids and, to a lesser extent, by their autoxidation [21,23,24,39]. These molecules, already detected in some watermelon cultivars, are characteristic of foods rich in fatty acids, such as, for example, avocado, fish oil and olive oil [39,46]. The C6 aldehydes hexanal and 2-hexenal play very important roles in the food and perfume industry. They impart a pleasant, extremely characteristic “green” and fresh fragrance, notes particularly appreciated and sought after [21]. In addition, some authors [32] suggest the possible formation of hexanal (although in small quantities) as a result of lycopene degradation, a carotenoid abundantly present in watermelon fruit. 

In the AM_WDF sample, hexanal and pentanal are particularly abundant and cover, respectively, 16% and 6.62% of the entire VOCs fraction TIC area. We can hypothesize a higher concentration of some polyunsaturated fatty acids in this sample. In particular, these two species are associated with the natural degradation of linoleic acid. This experimental observation may be related to the residual quantity of unripe white seeds inside the pomace. In fact, their germplasm oil may have changed the composition of the final product, the WDF.

The branched-chain molecules, 2-methylbutanal and 3-methylbutanal, described with a “malty-like” aroma, have been detected in all different samples analyzed pertaining to the three cultivars. These compounds are suggested to be synthesized from isoleucine and leucine, respectively, although the pathways are not well recognized yet [47]. However, it cannot be established which fraction of the total quantity of 2-methylbutanal and 3-methylbutanal is directly related to the aforementioned amino acids. However, these amino acids are probably indiscriminately present in the three cultivars. 

### 3.2. Alcohols

The trend of alcohols in the three cultivars is similar to that observed for the aldehydes. The AM_WDF sample presents the greatest abundance (as TIC area) of these molecules, while G_WDF is characterized by a higher number of detected analytes. In general, however, these compounds have a low TIC abundance and make a minor contribution to the fibers aroma if compared with the previously described aldehydes. Nevertheless, some exceptions to this trend can be detected and the *Crimson Sweet* sample is one of these.

1-Pentanol is the absolute most abundant alcohol found in the HS of *Asahi Miyako* fiber. It is one of the main products of linoleic acid degradation. This observation, together with the one made about pentanal and hexanal from the same fatty acid, would strengthen the hypothesis that linoleic acid is present to a greater extent in this cultivar. 

C9 alcohols are characteristic compounds of the aroma of numerous *Cucurbitaceae* fruits and vegetables [22,24,25,26,29]. However, C9 alcohols were not identified in the AM_WDF, nor were they in the G_WDF samples. In contrast, these analytes were detected in the HS of the CS_WDF sample. Among these, 3,6-nonadien-1-ol is one of the species most characterizing the aroma of *Citrullus lanatus* [37], because it has a strong scent of fresh watermelon and a low perception threshold level. 

We can affirm that *Gavina*^®®^ and *Crimson Sweet* cultivars show a composition of the HS and an aromatic profile in line with what is reported in the literature, although some differences are present. The first, in fact, is richer and characterized by C9 aldehydes, while the second is characterized by C9 alcohols. 

### 3.3. Ketones

As can be seen from Table 1, the CS_WDF sample is the one with the lowest number and quantity (TIC relative abundance) of ketones if compared with the other two samples. Acetoin (3-hydroxy-2-Butanone) has been detected in the three analyzed samples. In particular, the highest abundance is found in the HS of AM_WDF. This ketone has a pleasant creamy-yogurt smell, and is generally used in the food industry to enhance the flavor of some products [48,49]. It has been identified in several dietary products such as yogurt and cheese, vinegar, fruits, vegetables and some types of flours [49]. It also has good biocompatibility and solubility, extending its usage in soaps, lotions and cosmetics [48]. Acetoin has three natural origins: (i) microbial fermentation, (ii) vegetable synthesis by fruits or plants, and (iii) animal synthesis. Its biosynthesis in plant cells has been confirmed in the literature. It has also been reported that, along with other odorous compounds, it produces rich fragrances in plants with the purpose of attracting insects and other animals to help their pollination and propagation [49]. In our case, it is quite unlikely that fermentation processes took place markedly given the speed of handling the watermelon pulp from the fresh fruit to the dried fiber. Thus, most of acetoin could instead have a natural origin. 

The greatest difference found within this class of molecules concerns acetone. Its abundance is significantly higher in the CS_WDF and AM_WDF samples than in the G_WDF one. In some previous studies [50], the presence of acetone in the radical peroxidation mechanism of lipids has been ascertained, suggesting the measurement of this analyte as an indicator of the oxidative state of fatty substances. Therefore, we can hypothesize that the AM_WDF sample is richer in some fatty acids, the degradation of which can give rise to this volatile molecule. Considering the previous results related to aldehydes and alcohols, it seems that the AM_WDF sample has an aromatic profile scarcely influenced by both C9 aldehydes and alcohols. The most significant contribution is given by molecules produced mainly by the degradation of linolenic acid, especially hexanal, acetone and 1-pentanol. The poor separation of the pulp from the small immature white seeds in the fruits of the *Miyako* watermelon probably makes the fiber richer in certain lipid components. Although present in modest quantities, they could significantly contribute to the aromatic bouquet of the examined matrices.

### 3.4. Esters

Only two esters were identified in the HS of the analyzed samples. These experimental data are in line with those reported by several authors in the literature, which confirm the almost complete absence of esters in various *Citrullus lanatus* seedless cultivars [24,29,40]. The reason is related to the non-climacteric characteristic of watermelon [29]: it does not continue to ripen after being harvested, unlike climacteric fruits such as apples, bananas, peaches, etc. For the latter, ethylene is produced during ripening, and it activates biochemical and enzymatic mechanisms linked to the production of volatile esters. This observation is very interesting, because it shows that genetic factors play a fundamental role in the definition of some distinctive traits of the fruit considered, especially with regard to aroma, flavor, organoleptic characteristics, etc. Not surprisingly, although watermelon is a non-climacteric fruit, Beaulieu et al. [29] do not rule out the possibility that there may be some differences between seeded and seedless cultivars with regard to volatile ester content. Our results seem to confirm this hypothesis, albeit limited to the *Crimson Sweet* cultivar. In fact, ethyl acetate has been detected in significant quantities in the CS_WDF samples. 

### 3.5. Terpenes and Derivatives

It is well known that terpenes and terpenoids play a fundamental role in defining the aroma of plants, flowers, and fruits. The G_WDF sample is rich in monoterpenes and monoterpenoids such as limonene, terpinene, and its isomers, citronellol, borneol, terpineol, cubebene, and isopulegol, which are completely absent in the other two samples. Although these molecules follow similar biosynthetic pathways, they possess specific and unique fragrant characters. Limonene, one of the main aroma constituents of citrus family, has been found in various fruits, including watermelon [24,25,31]. It is a particularly sought-after molecule, appreciated in the food, cosmetic and soap industries, as it produces a very pleasant citrus and fresh fragrance. Linalool, typically present in the essential oils of lavender, citrus, jasmine, and in the essence of rosewood, imparts citrus and floral notes. This molecule undergoes epoxidation followed by rearrangement, which leads to the formation of trans-linalool oxide [51]. β-Terpinene, γ-Terpinene and α-Terpineol are cyclic monoterpenes that, although present in small quantities, help to enrich and complete the G_WDF aroma with floral, citrus, and woody notes.

Carotenoids belong to the terpene class, and they include innumerable sub-classes of metabolites, both primary and secondary. The primary ones include carotenoids, gibberellins, and phytosterol molecules, which are necessary for various cellular functions such as growth and maintenance. The secondary ones play roles mostly attributable to the interactions between the plant and the surrounding environment [52]. Carotenoids protect cells from oxidative stress induced by particularly reactive species such as free radicals, greatly contributing to the added value of foods they are contained in [44]. Furthermore, from a nutritional point of view, their intake is strongly recommended due to their strong antioxidant activity [40,44]. However, because of the high degree of unsaturation, these molecules are particularly sensitive to degradation induced by heat, biological oxidation processes or atmospheric oxygen, leading to the formation of characteristic volatile compounds [53]. 

Given the continuous increase in the demand for natural aromas, the flavor and fragrance industries are increasingly looking for new methods to obtain natural substances to replace their synthetic counterparts. Methods concerning the use and transformation of carotenoids are receiving increasing attention, as they represent a natural alternative that can provide valuable substances and sought-after aromas [43]. The detection of carotenoid degradation compounds in commercial products provides information on quality and acceptability, nutritional value, vitamin content, color, and visual impact [40]. From the identification of the VOCs in the HS of some fruits, it is possible to hypothesize the presence or absence of a given carotenoid. Through this rough estimate, conclusions can be drawn regarding the abundance of one carotenoid compared to another in different cultivars. 

Lycopene is accumulated in plant tissues and typically acts as a simple precursor for the biosynthesis of other carotenoids [40]. In some fruits, such as tomato and watermelon, it is instead present in high quantities, producing the distinctive red color. As shown in Figure 6, lycopene degradation gives rise to various molecules with lower molar masses, following various degradation pathways.

With the term “citral”, or 3,7-dimethyl-2,6-octadienal, we mean the cis- and trans- mixture of the neral and geranial acyclic monoterpenes. They impart a “citrus-like fresh and fruity” aroma. In nature, they are found mainly in citrus fruits and are particularly exploited in the food and cosmetic industries because of their characteristic and pleasant fragrance. In addition to watermelon, they are also present in some cultivars of apples, tomatoes, paprika and, more generally, in species containing high levels of lycopene or its precursors [22,30,40]. Therefore, α-citral and β-citral are considered markers of lycopene oxidation [22,30,32]. In addition to citral, we also have identified its epoxidation product, 2,3-epoxy-geranial, a compound already detected by other authors in similar matrices, which is formed by a mechanism that is easily established in contact with the atmospheric oxygen [32]. These degradation compounds have a higher TIC abundance in the HS of G_WDF. 

The most abundant species associated with lycopene degradation is 6-methyl-5- hepten-2-one. Its formation is linked to the oxidation or degradation of lycopene, α-farnesene, citral or conjugated trienols [22,29,30]. The highest TIC abundance is found in the AM_WDF sample, followed by the G_WDF one. The CS_WDF specimen is confirmed to be the least aromatic, also considering the terpene content.

β-Carotene is found in high concentrations in various vegetables such as carrots, sweet potatoes, spinach, and pumpkin. It is the precursor of vitamin A [44]; therefore, it has a marked nutritional importance, and its intake is particularly recommended. The species shown in Figure 7 are commonly related to the degradation of β-carotene [30,43]. The most representative degradation compounds are β-ionone, produced following the breaking of the C9-C10 bond, and dihydroactinidiolide, which is hypothesized to originate from 5,6-ionone epoxide through a radical oxidation reaction [44] or from the oxidation of C8-C9 bond [43]. These degradation products have a particularly low odor threshold. Consequently, although β-Carotene is not the main carotenoid in watermelon pomace, it significantly contributes to its aroma. β-Ionone imparts a floral, slightly fruity and pleasant aroma [44]; it is widely exploited in perfumery production and is used for the synthesis of vitamin A [21]. β-Ionone has been detected in its epoxidized form, 5,6-Ionone epoxide. Epoxidation of these terpenes occurs quite easily in the presence of atmospheric oxygen. These two species are the most abundant among those related to β-carotene degradation: probably the C9-C10 bond is the most susceptible to breaking. These degradation products are more abundant in the HS of the G_WDF specimen, followed by the AM_WDF one. CS_WDF sample analysis only shows traces of β-Ionone. A peculiarity of the *Asahi Miyako* sample is the presence of α-ionone, although in small quantities. This species originates primarily from α-carotene, isomer of β-carotene, with similar nutritional properties, including the activity as a precursor of vitamin A [40]. α-ionone has very similar characteristics to its β-isomer: it has a low threshold, and a pleasant floral and fruity aroma [40,43].

### 3.6. Other Compounds

In this heterogeneous group, we collected the residual analytes, which apparently do not seem to be related to the compounds identified and previously classified according to the molecular structure and functional groups. The main differences between the three samples were the presence of 2-methyl-furan and 2-pentyl-furan. They are particularly abundant in the *Miyako* fiber and scarcely present in the *Crimson Sweet* one. 2-pentyl-furan is associated with the radical peroxidation of fatty acids [21,27], and its high abundance in the AM_WDF sample is consistent with previous observations, i.e., the aromatic profile of this cultivar fiber is particularly influenced by lipid peroxidation processes.

One-way analysis of variance (ANOVA) was performed to assess whether there was a statistically significant difference among the G_WDF, AM_WDF, and CS_WDF samples according to the VOCs classes (Table 1).

The fibers obtained from the three cultivars of *Citrullus lanatus* present a significantly different VOC composition, with a *p*-value, for all the investigated compounds, smaller than the significance level (0.05). In particular, specific considerations can be developed for some classes of molecules listed in Table 2.

(*i*) The G_WDF samples show an aromatic bouquet defined mainly by the class of terpenes (TER). It presents the most complex and characteristic fragrance, since terpenes are extremely aromatic, conferring pleasantness with strong effects on the perception and appreciation of the consumers. Some terpenes can be attributed to lycopene degradation. At present, we are unable to establish whether their origin is strictly related to the presence of significant quantities of the precursor compound or the biosynthetic pathways from which they can be generated. It also shows the higher level of C9 aldehydes in terms of both concentration and number. These compounds are the most characteristic of the aroma of the *Cucurbitaceae* fruits.

(*ii*) M_WDF specimens show the highest percentage of aldehydes. They could originate from metabolic pathways or fatty acid degradation. The latter hypothesis is reinforced by the presence of other species, including alcohols and ketones, which are associated with the oxidation of some polyunsaturated fatty acids. It should be emphasized that the pulp of the processed fruits of this cultivar contained small white seeds, which are difficult to separate from the pomace itself. Consequently, it is probable that a small fraction remained in the finished product, with the consequent modification of its composition.

(*iii*) CS_WDF samples have the highest ester concentration. This observation is consistent with the result of several previous studies [24,25,28,29]. Overall, the fiber of this cultivar has a leaner and simpler aromatic bouquet. The abundance of the characteristic watermelon VOCs is low, as well as that of terpenes.

## 4. Materials and Methods

### 4.1. Samples Preparation

*Gavina*^®®^, *Asahi Miyako* and *Crimson Sweet* watermelons, produced in 2020, were partly purchased at some local supermarkets in the city of Modena, and partly purchased directly from various producers in the Modena area.

The degree of ripeness of the fruits was the right one and typical for consumption. The watermelons were readily used in the condition they were in at the time of purchase. Five different fruits were selected for each cultivar in order to provide a representative analytical sample.

The fruits were carefully washed with water, peeled, and deseeded. Three slices of about 100 g each were taken from different orientations of each fruit and used for one composite sample. The red pulp was shredded by a kitchen mixer to obtain a homogeneous slurry-meal (Figure 8).

The pulp was then filtered to separate juice from fibrous fraction, which was finally vacuum freeze-dried (CHRIEST, Mod. Alpha 1–2 LDplus; Direct Industry, Germany) for 20 h, 15 of which were at 1 mbar, and the remaining at 0.001 mbar, to obtain the whole meal, raw fiber (WDF). The resulting fibers were immediately homogenized in a grinding mill equipped with a rotor made in Ti and a sieve of 500 μm, maintaining the equipment at low temperature by refrigerating with some few drops of liquid N_2_. This strategy allowed to preserve as much as possible the aromatic characteristics of the fibers. Then, a consistent set of samples was prepared as follows: about 0.5 g of material were transferred in 10 mL glass vials which were sealed tightly with Teflon/silicone septa.

The vacuum freeze-dried specimens were immediately analyzed after they have been prepared to characterize the VOC’s aromatic fraction.

All the analyses were carried out at least in triplicate, as described in the following sections.

### 4.2. Volatile Organic Compounds Sampling—HS SPME

A Solid Phase Micro-Extraction (SPME) holder (Supelco Inc., Bellefonte, PA, USA) was used to manually perform the SPME HeadSpace (HS) analysis.

Before being analyzed, all the samples were sonicated for 30 min in a thermostatic bath at 40.0 ± 0.1 °C to favor the transfer of volatile compounds from the matrix to the headspace. After this step, the extraction of volatiles was performed by manually exposing a 2 cm long SPME fiber composed of CW/DVB/PDMSO (Supelco, Bellefonte, PA, USA) to the HS of the vial for 15 min at the same temperature. Finally, the fiber was withdrawn and inserted into the injector port of a GC-MS system for desorption of the analytes at 250 °C for 15 min. 

Reproducibility of experimental procedures has been established working with at least three replicate samples of the same matrix and collecting many different measures for each vial.

Some blank tests corresponding to the analysis of an external standard solution containing 1-decanol (conc. 150 µg/g ethanolic solution) were performed periodically after a certain number of chromatographic runs (5) relating to real samples. 

### 4.3. GC-MS Analysis

GC-MS analysis of the extracted volatile compounds was performed on Agilent 6890N Network gas chromatograph system coupled with a 5973N mass spectrometer (Agilent Technologies, CA, USA). A DB-5MS UI column (60 m × 0.25 mm i.d., 1.00 μm film thickness; J&W Scientific, Folsom, CA, USA) was used and the carrier gas was Helium erogated at a flow rate of 1.0 mL/min. The SPME injections were performed in splitless mode and the detector started to operate immediately after each injection. The column temperature was held at 40 °C for 5 min, then increased to 160 °C at a rate of 10 °C/min and to 270 °C at 8 °C/min and held for 5 min. The transfer line was heated to 270 °C.

The mass spectrometer operated in electron impact (EI) ionization mode at 70 eV, in full scan acquisition mode, with a *m*/*z* scanning range from 25 to 300.

Chromatograms and mass spectra were analyzed using the Enhanced ChemStation software (Agilent Technologies, CA, USA). Putative identification of volatile compounds was achieved by matching the mass spectra with the data system library (NIST14/NIST05/WILEY275/NBS75K) and by using some databases accessible via the web such as National Institute for Standards and Technology (NIST database https://webbook.nist.gov accessed on 12 October 2022) and Mass Bank of North America (https://mona.fiehnlab.ucdavis.edu accessed on 7 October 2022).

Linear Retention Index (LRI) was used for an additional comparison between our data and those reported in the literature and in the NIST Standard Reference Database, considering only values referring to analyses carried out under the same operative conditions (instrumental specifications and heating ramp).

LRIs of compounds have been calculated from a series of n-alkanes (C6, C9, C12, C14, C16) subjected to the same analysis procedure adopted for the samples. The latter proved to be particularly useful for the distinction of E/Z isomers, since these species produce mass spectra that are difficult to differentiate.

Finally, some analytes were identified by comparing their mass spectra with those of their respective pure standards (when available), analyzed by HS-SPME-GC-MS under the same operating conditions used for the samples.

The volatile compounds such as silane and siloxane derivatives or volatile organic compounds related to the sorbent fiber were discarded and are not reported in the GC-MS output tables. The estimation of the amount of each volatile identified in the SPME-GC-MS analysis was expressed as the Total Ion Current (TIC) peak area.

All the data shown in the tables relate to values obtained from analysis performed at least in triplicate. The reproducibility of the results was expressed as standard deviation in the tables. When absent, the apex ^a^ indicate that SD < 0.05.

Procedures for HS-SPME-GC-MS measurements have been proposed and applied in our previous studies [5,54] on the aroma profiles of similar plant-based samples.

### 4.4. Chemicals and Reagents

The chemicals used during the study are the following: (i) 2-methyl-1-propanol; 1-pentanol; 2-ethyl-1-hexanol; phenylethyl alcohol; 2-phenoxyethanol; 3-methylbutanal; 2-ethyl-2-butenal; ethyl butanoate; ethyl-3-hydroxybutanoate; 3-hydroxy-2-butanone (acetoin); 2- methylbutanoic acid, and benzothiazole were obtained from Sigma—Aldrich product, distributed by Merck KGaA, Darmstadt, Germany. (ii) decanal; methyl acetate; 1-decanol; n-hexane; nonane; dodecane; tetradecane, and hexadecane were obtained from Carlo Erba Reagents, Milano (Italy).

### 4.5. Statistical Analysis

Experimental data were compared by applying analysis of variance (one-way ANOVA), by running in the Matlab^®®^ 2020b environment (The Mathworks Inc., Natick, MA, USA). The level of significance was determined at *p* < 0.05 to see whether there are statistical differences between the mean values.

## 5. Conclusions

The goal of this work was to develop an analytical procedure for the characterization of the volatile profile of WDF obtained from different *Citrullus lanatus* cultivars. For this purpose, the WDF samples were analyzed by means of HS-SPME/GC-MS.

The different groups of identified analytes allowed to discriminate WDF samples according to their cultivar. Furthermore, it was possible to highlight that G_WDF sample has a more complex aromatic bouquet characterized by many different types of terpenes. The AM_WDF sample shows a volatile fraction more linked to the degradation of polyunsaturated fatty acids, probably attributable to contamination of the sample by unripe white seeds, rich in germplasm oil. The CS_WDF sample has a leaner and simpler aroma, and is characterized by fewer analytes.

The obtained results could be particularly useful in identifying some target compounds as qualitative and quantitative markers, and can increase knowledge on the influence of cultivars on the final composition of *Citrullus lanatus* fibers. The use of HS-SPME/GC-MS approach also provided detection of analytes present in low amounts, enabling effective screening of a wide variety of volatile and semi-volatile compounds. A whole chemical characterization of the aroma profile of WDF is very important to gain information on the quality of the obtained fibers and to promote their use in the food industry, supporting the recovery of watermelon by-products as well.

## Figures and Tables

**Figure 1 molecules-27-08747-f001:**
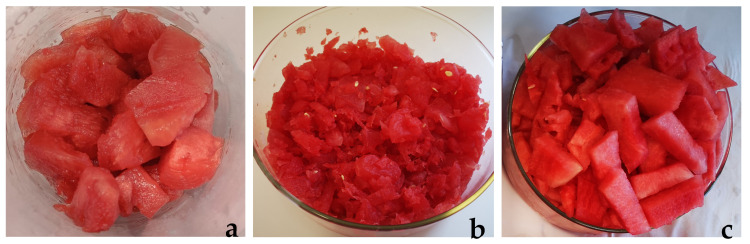
Images of the three watermelon pulps, seeds removed, of (**a**) *Gavina*^®®^; (**b**) *Asahi Miyako*; (**c**) *Crimson Sweet*.

**Figure 2 molecules-27-08747-f002:**
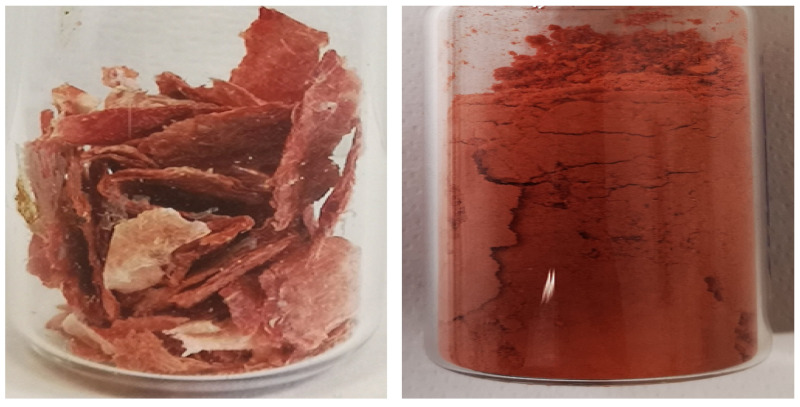
Images of the G_WDF sample: flakes immediately after the drying process (**left**), and powder after the subsequent grinding process (**right**).

**Figure 3 molecules-27-08747-f003:**
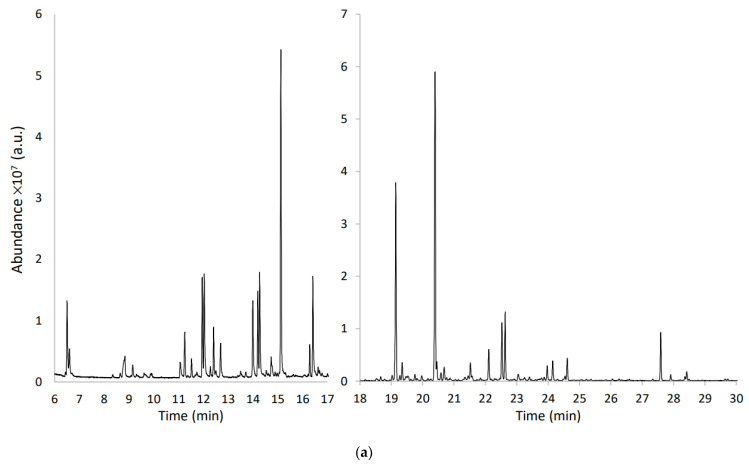
(**a**) TIC (Total Ion Current) chromatogram of the VOCs from G_WDF fibrous sample obtained by HS-SPME-GC-MS (a.u. = arbitrary units). (**b**) TIC chromatogram of the VOCs from AM_WDF fibrous sample obtained by HS-SPME-GC-MS (a.u. = arbitrary units). (**c**) TIC chromatogram of the VOCs from CS_WDF fibrous sample obtained by HS-SPME-GC-MS (a.u. = arbitrary units).

**Figure 4 molecules-27-08747-f004:**
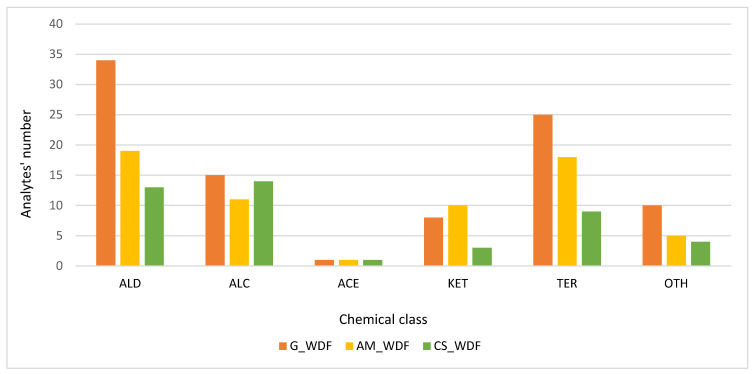
Histogram representation of the number of analytes by chemical class for each examined WDF sample. ALD (aldehydes), ALC (alcohols), ACE (acetate esters), KET (ketones), TER (terpenes and their derivatives), OTH (other analytes); G = *Gavina*^®®^, AM = *Asahi Myiako*, CS = *Crimson Sweet*.

**Figure 5 molecules-27-08747-f005:**
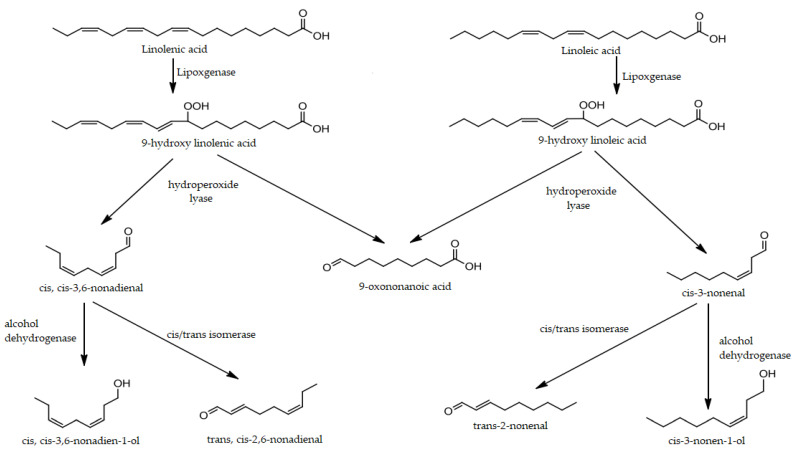
Proposed biosynthetic pathway for watermelon volatiles and their counterparts.

**Figure 6 molecules-27-08747-f006:**
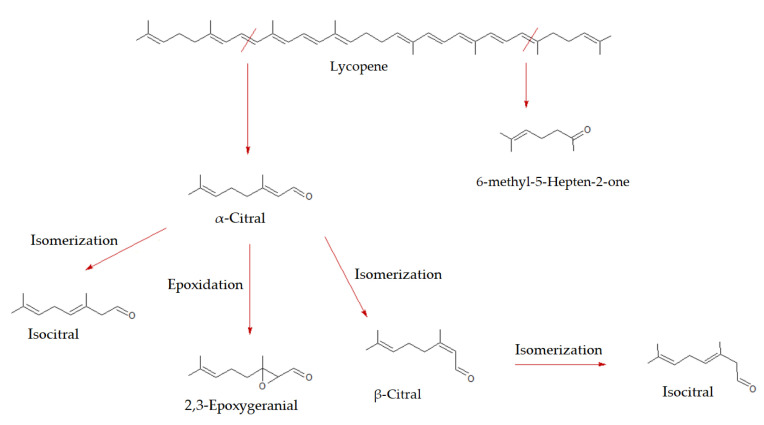
Proposed biosynthetic pathway for watermelon volatiles and their counterpart: lycopene degradation.

**Figure 7 molecules-27-08747-f007:**
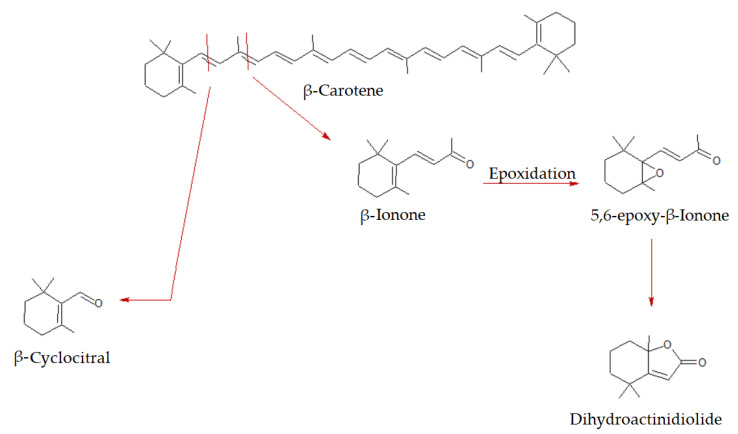
Proposed biosynthetic pathway for watermelon volatiles and their counterparts: β-carotene degradation.

**Figure 8 molecules-27-08747-f008:**
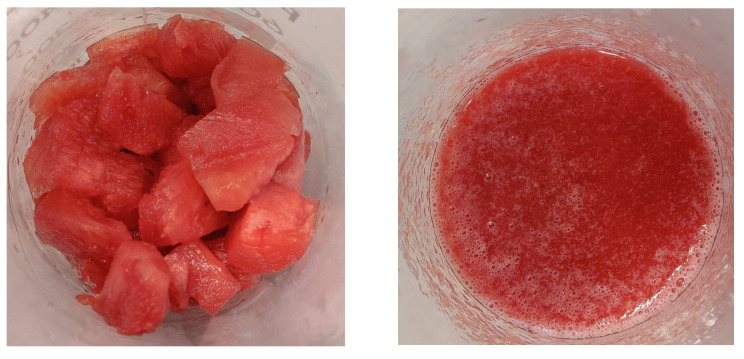
Watermelon (*Citrullus lanatus*) variety *Gavina*^®®^ pomace: natural (**left**) and homogenized (**right**).

**Table 1 molecules-27-08747-t001:** VOC composition of vacuum-freeze-dried G_WDF, AM_WDF and CS_WDF identified through HS-SPME-GC-MS analysis, grouped by chemical classes. *^a^* Data are expressed as mean (n = 3) TIC area 10^7^ ± SD_(3)_.

Compound	LRI	^#^ ID	Aroma	Area (G_WDF)	Area (AM_WDF)	Area (CS_WDF)	Ref.
*Alcohols*							
ethanol	466	A, B	Alcoholic	2.2 ± 0.1	-	1.4 ± 0.1	[25,26,27,28]
2-methyl-3-buten-2-ol	613	A, B	Grass, earthy, oily	0.18	-	-	-
1-penten-3-ol	686	A, B	Pungent, green, fruity	3.6 ± 0.1	7.5 ± 0.2	1.1	[24,26,27,28,29]
3-methyl-1-butanol	736	A, B	Alcoholic	-	-	0.32	-
2-methyl-1-butanol	741	A, B	Fatty, wine, cocoa	0.36	3.1 ± 0.1	6.2 ± 0.2	[24,25,27,28]
1-pentanol	765	A, B, C	Pungent, fermented	3.0 ± 0.1	43.4 ± 0.5	1.7 ± 0.1	[24,25,26,27,30,31,32]
(Z)-2-penten-1-ol	768	A, B, D	Green, phenolic, ethereal	4.2 ± 0.2	-	-	[26,27,28,29,30]
2,3-butanediol	777	A, B	Creamy, fruity, buttery	0.42	1.1 ± 0.1	0.17	-
2,5-dimethyl-1,5-hexadien-3-ol	839	A, B	-	1.1 ± 0.1	4.3 ± 0.1	0.18	-
1-hexanol	851	A, B	Green, flowery	0.37	1.3 ± 0.1	0.45	[22,24,25,27,28,29,30,31]
1-octen-3-ol	936	A, B	Mushroom, earthy, green	2.4 ± 0.1	2.2 ± 0.1	0.20	[25,27,28,29,30,31]
benzyl alcohol	985	A, B	Sweet, floral, fruity		3.3 ± 0.1	0.10	[25,27,29]
1-octanol	997	A,B	Waxy, green, citrus, floral	1.5 ± 0.1	-	-	[24,26,27,28,29,30]
2,6-dimethyl-cyclohexanol	1044	A, B	-	13.2 ± 0.3	11.5 ± 0.3	1.5 ± 0.1	-
2,6-nonadien-1-ol	1051	A, B	Cucumber, green	1.0 ± 0.1	-	-	[27,28,30,33]
(Z,Z)-3,6-nonadien-1-ol	1056	A, B, D	Watermelon, fresh, melon	-	-	0.72	[22,24,25,26,28,33,34,35]
(Z)-6-nonen-1-ol	1056	A, B	Green, melon, cucumber	1.1	-	-	[24,25,26,27,28,29,30,33]
(Z)-3-nonen-1-ol	1058	A, B	Fresh, waxy, green, melon	-	0.50	2.3 ± 0.1	[22,24,25,26,28,33,34,35]
1-nonanol	1061	A, B	Fresh, clean, floral, grass	-	-	0.68	[22,24,25,26,27,28,33,35]
3,4-dimethylcyclohexanol	1131	A, B	-	2.2 ± 0.1	1.5 ± 0.1	-	-
*Aldehydes*							
acetaldehyde	435	A, B	Pungent, fresh, penetrating	5.5 ± 0.2	-	-	[22,25,26,29,30,36]
propanal	498	A, B	Ethereal, pungent, fruity	1.3 ± 0.1	-	-	[25,36]
2-methyl-propanal	559	A, B	Fresh, aldehydic, floral	0.16	1.2 ± 0.1	-	-
(Z)-2-butenal	570	A, B, D	-	0.23	2.6 ± 0.1	-	-
butanal	593	A, B	Pungent, malty, green	0.21	-	-	-
(E)-2-butenal	655	A, B, D	Green, fruity	0.82	2.5 ± 0.1	0.06	[24,25,29,30]
3-methyl-butanal	661	A, B	Ethereal, aldehydic, peach	1.7 ± 0.1	23 ± 0.4	2.5 ± 0.1	[24,31,37]
2-methyl-butanal	671	A, B	Chocolate, nutty, furfural	0.72	16 ± 0.3	1.5 ± 0.1	[24,31]
pentanal	703	A, B	Fermented, fruity, berry	1.9 ± 0.1	35 ± 0.5	1.5 ± 0.1	[24,25,26,27,28,29,31]
2-methyl-2-butanal	748	A, B	Pungent, green, penetrating	0.28	-	-	-
(E)-2-pentanal	758	A, B, D	Pungent, green, fruity, apple	3.4 ± 0.1	2.6 ± 0.1	-	[24,25,26,27,29,30,31,36]
3-methyl-2-butanal	784	A, B	Sweet, fruity, pungent, nut	0.93	2.0 ± 0.1	0.29	-
hexanal	798	A, B	Green, fatty, fruity, woody	12 ± 0.3	85 ± 0.8	6.0 ± 0.2	[22,24,25,27,28,30]
(Z)-2-hexenal	837	A, B, D	Green	0.11	-	-	[24]
(E)-2-hexenal	843	A, B, D	Green, almond, fruity	4.2 ± 0.1	3.4 ± 0.1	0.63	[24,25,27,29,30,36]
4-heptenal	878	A, B	-	0.32	0.54	-	[25,26,29,30]
heptanal	880	A, B	Fresh, fatty, green, grass	1.2 ± 0.1	3.0 ± 0.1	0.20	[24,25,26,29,30,31,37]
(E,E)-2,4-hexadienal	890	A, B, D	Green, fruity, citrus, waxy	0.34	0.38	-	[29,31]
(E)-2-heptenal	923	A, B, D	Green, fatty, oily, fruity	1.9 ± 0.1	2.6 ± 0.1	0.12	[24,25,26,27,29,30]
octanal	954	A, B	Orange, waxy, peel, green	0.73	0.45	0.13	[24,25,27,28,29,30,31]
(E,E)-2,4-heptadienal	963	A, B, D	Fatty, sweet, melon, spicy	1.1 ± 0.1	0.53	-	[24,25,27,29,30,31]
benzenacetaldehyde	994	A, B	Honey, sweet, rose, floral	6.5 ± 0.2	-	0.21	[25,31]
(E)-2-octenal	995	A, B, D	Green, nut, fat	-	1.6 ± 0.1	-	[24,25,27,29,30,31,33]
(E)-4-nonenal	1017	A, B, D	Fruity	1.8 ± 0.1	-	-	[24,29,30,31]
nonanal	1023	A, B	Citrus, lemon, cucumber	7.4 ± 0.2	-	0.70	[22,24,27,28,29,31,37]
(Z,Z)-3,6-nonadienal	1046	A, B, D	Fresh-cut watermelon	0.79	-	-	[25,26,27,30,34,37]
(Z)-2-nonenal	1052	A, B, D	Cucumber, fatty, waxy	0.69	-	-	[24,26,30,31,36,37]
(E,E)-2,6-nonadienal	1054	A, B, D	Melon, cucumber, fatty	0.45	-	-	[24,26,31]
(E,Z)-2,6-nonadienal	1059	A, B, D	Cucumber, melon, green	24 ± 0.4	0.74	-	[22,26,27,29,30,31,37]
(E)-2-nonenal	1062	A, B, D	Fatty, green, melon, waxy	27 ± 0.4	2.3 ± 0.1	0.79	[22,24,25,26,29,30,36]
decanal	1090	A, B	Zest, waxy, orange, floral	2.0 ± 0.1	-	-	[24,25,31]
(E,E)-2,4-nonadienal	1101	A, B, D	Cucumber, melon, waxy	0.66	0.15	-	[24,25,29,30,37]
(E)-2-decenal	1127	A, B, D	Waxy, earthy, coriander	0.87	-	-	[24,30,31,33]
(E,Z)-2,4-decadienal	1165	A, B, D	Fatty, orange, citrus, fresh	0.47	-	-	[24,25,27,30]
dodecanal	1214	A, B	Soapy, waxy, citrus, floral	0.24	-	-	[24]
*Ketones*							
acetone	495	A, B	Solvent, ethereal, apple, pear	3.1 ± 0.1	55 ± 0.5	8.5 ± 0.2	-
3-methyl-2-butanone	588	A, B	Camphor	0.61	3.1 ± 0.1	-	-
2-butanone	597	A, B	Ethereal, fruity, camphor	-	4.4 ± 0.1	-	-
1-penten-3-one	689	A, B	Pungent, pepper, garlic, onion	4.5 ± 0.1	-	-	[25,26,29,30,38]
2,3-pentanedione	698	A, B	Buttery, nutty, caramelized	0.42	-	-	[25,29]
acetoin	713	A, B	Sweet, buttery, creamy, milky	1.5 ± 0.1	11 ± 0.3	0.61	[31]
1-(1-cyclohexen-1-yl)-ethanone	869	A, B	-	0.66	-	-	-
2-heptanone	871	A, B	Fruity, spicy, sweet, grass	-	0.78	-	[27]
2-methyl-1-hepten-6-one	929	A, B	-	-	2.1 ± 0.1	-	-
3-octen-2-one	979	A, B	Earthy, oily, sweet	-	2.2 ± 0.1	-	[29]
(E,Z)-3,5-octadien-2-one	1002	A, B, D	Fatty, fruity, hay, green	1.4 ± 0.1	1.7 ± 0.1	0.06	[25,39]
acetophenone	1013	A, B	-	-	0.17	-	-
(E,E)-3,5-octadien-2-one	1019	A, B, D	Fruity, green, grassy	0.62	1.3 ± 0.1	-	[25]
*Esters*							
ethyl acetate	612	A, B	Fruity, pineapple, apricot	-	-	24 ± 0.4	[22,25]
3-methylcyclopentyl acetate	883	A, B	-	5.1 ± 0.2	4.9 ± 0.1	-	-
*Terpenes and derivatives*							
6-methyl-5-hepten-2-one	940	A, B	Fruity, apple, must, creamy	81 ± 0.6	120 ± 0.7	10 ± 0.3	[22,24,25,26,28,30,40]
6-methyl-5-hepten-2-ol	946	A, B	Sweet, coriander, green, oily	-	0.53	0.08	[24,26,29,32]
β-terpinene	944	A, B	-	2.2 ± 0.1	-	-	[41]
γ-terpinene	970	A, B	Terpenic, sweet, citrus	0.25	-	-	[41]
cymene	979	A, B	Bitter, woody, citrus	1.0	-	-	[42]
limonene	984	A, B	Citrus, orange, sweet, fresh	133 ± 0.9	-	-	[24,25,31,41]
citronellol	990	A, B	Floral, rose, sweet, citrus	3.8 ± 0.1	-	-	-
linalool oxide	1010	A, B	Woody, floral, fresh	0.52	0.30	-	[41]
linalool	1021	A, B	Citrus, orange, floral, waxy	2.7 ± 0.1	-	-	[41]
6-methyl-3,5-heptadien-2-one	1025	A, B	Sweet, green, spicy, fresh	2.3 ± 0.1	1.6 ± 0.1	0.16	-
4-terpineol	1035	A, B	Woody, citrus, spicy	-	0.27	-	-
cymen-3-ol	1041	A, B	Spicy, phenolic, camphor	-	0.52	-	-
α-cyclocitral	1051	A, B	Saffron, tropical, grass, fruity	-	0.35	-	-
isocitral	1073	A, B	-	0.81	-	-	-
borneol	1084	A, B	Balsamic, woody, camphor	1.8 ± 0.1	-	-	-
α-terpineol	1097	A, B	Pine, woody, resinous, citrus	0.58	-	-	[41]
5-isopropenyl-2-methyl cyclo-pent-1-enecarboxaldehyde	1106	A, B	-	1.7 ± 0.1	0.25	0.25	-
2,3-epoxygeranial	1110	A, B	-	6.1 ± 0.2	1.1 ± 0.1	-	[32,40]
β-cyclocitral	1113	A, B	Saffron, tropical, grass, fruity	-	2.6 ± 0.1	0.29	[23,24,25,26,40,41,42,43,44]
β-citral	1116	A, B	Fresh, lemon, sweet, green	8.5 ± 0.2	-	-	[23,24,25,27,28,29,40,44]
1,4-dimethyl-3-cyclohexen-1-carboxaldehyde	1121	A, B	-	0.76	0.14	-	-
α-citral	1132	A, B	Tea, fresh, mint, fruity	9.0 ± 0.2	0.38	0.12	[24,27,28,29,32,33,40,43]
β-homocyclocitral	1140	A, B	Camphor, fresh, woody	0.47	0.92	0.07	-
isopulegol	1154	A, B	Mint, fresh, grass	0.83	-	-	-
isocitral	1180	A, B	-	0.22	-	-	-
cubebene	1217	A, B	Grass, waxy	0.14	-	-	-
α-Ionone	1236	A, B	Sweet, wood, floral, fruity	-	0.22	-	[24]
geranylacetone	1237	A, B	Rose, fresh, leaf, floral, green	18 ± 0.4	0.73	0.11	[23,24,25,26,27,28,29,30,40]
β-ionone	1266	A, B	Sweet, wood, violet, floral	3.6 ± 0.1	1.6 ± 0.1	0.08	[23,24,25,26,33,43,44]
5,6-β-ionone epoxide	1269	A, B	Fruity, sweet, wood, violet	0.58	0.49	-	[23,27,28,33,41]
dihydroactinidiolide	1309	A, B	Apricot, red, fruit, wood	0.76	0.84	-	[40,41,42,43]
*Others*							
2-methyl furan	604	A, B	Ethereal, vinegar, chocolate	0.18	9.0 ± 0.2	-	-
3-methyl furan	615	A, B	-	0.20	-	-	-
2-ethyl furan	706	A, B	Sweet, malty, earthy	0.35	-	-	[29]
dimethyl disulfide	755	A, B	Cabbage, vegetable, sulphur	-	-	0.21	-
furfural	828	A, B	Sweet, bread, caramel	-	-	0.32	[30]
2-n-butyl furan	875	A, B	Faint, fruity, sweet	-	0.80	-	-
methional	887	A, B	Tomato, potato, yeast	0.78	-	-	-
2,4-dimethyl phenol	905	A, B	Dark, roast, smoked	0.49	0.88	0.10	-
2-pentyl furan	947	A, B	Green, waxy, caramel	7.6 ± 0.2	19 ± 0.4	0.45	[21,24,25,27,31]
2-(2-pentenyl) furan	953	A, B	-	2.3 ± 0.1	0.69	-	[30]
5-hydroxy-2-methyl-3-hexenoic acid	1193	A, B	-	0.44	-	-	-
5-heptildihydro-2(3H)-furanone	1196	A, B	-	0.22	-	-	-
2-methyl propanoic acid, 3-hydroxy-2,2,4-trimethyl pentyl ester	1202	A, B	-	0.59	-	-	-

^#^ Identification is provided by: (A) mass spectral data of the libraries supplied with the operating system of the GC-MS and from mass spectra databases; (B) mass spectra found in the literature; (C) mass spectra and retention time of an injected standard; (D) LRI values, typically used for the identification of isomers. *^a^* SD < 0.05, unless otherwise noted.

**Table 2 molecules-27-08747-t002:** Compounds classes ^#^ identified in the HS-SPME-GC-MS analysis * of the fiber samples obtained from watermelon pomace.

Compound Class	G_WDF	AM_WDF	CS_WDF	*p*-Value
	$\underline{x}$ ± SD_(3)_	$\underline{x}$ ± SD_(3)_	$\underline{x}$ ± SD_(3)_	
ALC (alcohols)	7.8 ± 0.8	15.0 ± 1.6	15.3 ± 1.7	<0.05
ALD (aldehydes)	23.3 ± 1.8	35.1 ± 2.7	19.8 ± 1.6	<0.05
KET (ketones)	2.7 ± 0.5	15.5 ± 1.0	12.3 ± 0.8	<0.05
ACE (acetate esters)	1.1 ± 0.2	0.9 ± 0.1	32.7 ± 2.9	<0.05
TER (terpenes)	59.2 ± 3.1	25 ± 1.9	15.3 ± 1.7	<0.05
OTH (other compounds)	2.8 ± 0.4	5.7 ± 0.5	1.5 ± 0.3	<0.05

^#^ Data are expressed as mean % of each class to the total normalized peak areas on the basis of the sum of the TIC area. * mean of three replicates of the chromatograms, ± standard deviation SD_(3)_.

## Data Availability

Not applicable.

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
