# Peer review of "VOCs Analysis of Three Different Cultivars of Watermelon (Citrullus lanatus L.) Whole Dietary Fiber"

_molecules, 2022, doi:10.3390/molecules27248747_

Round 1

Reviewer 1 Report

The work proposed by Maletti et. to the. concerns the analysis of volatile organic compounds (VOCs) present in three different watermelon cultivars. a GC-MS analysis of the extracts was performed and the different identified components were compared using a statistical approach which identified differences between these three cultivars. The samples before the GC-MS analysis were extracted using the SPME head-space technique which allows saving organic solvents and above all not polluting the sample and the results that are presented. In my opinion the work deserves to be published in its current form since I have not found any criticism.

The authors should better explain if experiments have been carried out in the SPME phase to identify the best experimental conditions as well as in the chromatographic part, they should explain if they have used an already published method or have developed their own method. 

Author Response

OK, this has been done. We have added a sentence (in blue ink) at the end of paragraph 4.2, at page 20.

Reviewer 2 Report

Dear authors,

Your manuscript is well written and the topic is very interesting. The Introduction, as well as the Discussion section provide enough information so reader can understand the results and the main aim of the study. The text is clear and easy to understand.

However, I have several suggestions that might improve your manuscript even more:

- page headers and footers: please check the page number and the rest of the text in page headers and footers according to the MDPI template. The page numbers after page 5 are not correct.

- in the Abstract: I recommend providing one or two sentences at the end of the abstract describing the main results (for example: the results showed that cultivars can be differentiate according to their WDF VOCs content because...or something like that), instead of just listing the analyte groups.

- the names of compounds and compound groups: please write them in a unique way; in one part of the manuscript they are written with capital letter, and in the other part this is not the case. I recommend using lower case.

- in the Introduction: please write the percentage sign next to the number without a space (21%, 17% etc.) as you did in the rest of the text.

- in Table 1: is the (3) necessary after SD in the table caption? What does it represent? It is also mentioned in Table 2. Reconsider to split this large Table 1 into several smaller tables (aldehydes, alcohols...) to fit on one page, so it can be easier to read.

in Figure 4: please provide abbreviations descriptions in figure caption (ALD, ALC, AM, G, CS....etc.).

in Discussion: in paragraph where carotenes are described, the abbreviation C_WDF....should it be CS_WDF? Check through text.

in Table 2: should S be SD?

in Discussion: in the last part, under iii) you mentioned in second sentence that it is consistent with several previous studies. Please provide references for these studies on the end of this sentence.

Otherwise, it is my opinion that this manuscript is well written with a lot of relevant information that are supported with the given results.

Author Response

page headers and footers: please check the page number and the rest of the text in page headers and footers according to the MDPI template. The page numbers after page 5 are not correct.        

We thank the referee for the report, but we believe that this aspect can be improved by the editorial staff during the layout phase of the journal.

in the Abstract: I recommend providing one or two sentences at the end of the abstract describing the main results (for example: the results showed that cultivars can be differentiate according to their WDF VOCs content because...or something like that), instead of just listing the analyte groups.  

OK, this has been done. We have deleted the list of analyte groups and added three short sentences with some explanations.

the names of compounds and compound groups: please write them in a unique way; in one part of the manuscript they are written with capital letter, and in the other part this is not the case. I recommend using lower case. 

OK, this has been done.

- in the Introduction: please write the percentage sign next to the number without a space (21%, 17% etc.) as you did in the rest of the text.

OK, this has been done.

in Table 1: is the (3) necessary after SD in the table caption? What does it represent? It is also mentioned in Table 2.

SD(3) : this should be the correct formalism to represent the Standard Deviation with the number of replicates.

 Reconsider to split this large Table 1 into several smaller tables (aldehydes, alcohols...) to fit on one page, so it can be easier to read. 

For other articles in this Journal, we have been asked not to split the tables. However, if the editors agree, the analyte classes will be quickly separated into smaller subtables.

in Figure 4: please provide abbreviations descriptions in figure caption (ALD, ALC, AM, G, CS....etc.).

OK, this has been done.

in Discussion: in paragraph where carotenes are described, the abbreviation C_WDF....should it      be CS_WDF? Check through text.  

YES, a mistake occurred, and now has been corrected.

in Table 2: should S be SD?   

YES,  a mistake occurred, and now has been corrected.

in Discussion: in the last part, under iii) you mentioned in second sentence that it is consistent with several previous studies. Please provide references for these studies on the end of this sentence.   

OK, this has been done.

Otherwise, it is my opinion that this manuscript is well written with a lot of relevant information that are supported with the given results.